# Towards Robust Object Detection Invariant to Real-World Domain Shifts

**Qi Fan**[1]**, Mattia Segu**[2,3]**, Yu-Wing Tai**[1]**, Fisher Yu**[2]**, Chi-Keung Tang**[1]**,
Bernt Schiele**[3]**, Dengxin Dai**[3]
[1] The Hong Kong University of Science and Technology,
[2] ETH Zurich, [3] Max Planck Institute for Informatics, Saarland Informatics Campus

## Abstract

Safety-critical applications such as autonomous driving require robust object detection invariant to real-world domain shifts. Such shifts can be regarded as different domain styles, which can vary substantially due to environment changes, but deep models only know the training domain style. Such domain style gap impedes object detection generalization on diverse real-world domains. Existing classification domain generalization (DG) methods cannot effectively solve the robust object detection problem, because they either rely on multiple source domains with large style variance or destroy the content structures of the original images. In this paper, we analyze and investigate effective solutions to overcome domain style overfitting for robust object detection without the above shortcomings. Our method, dubbed as Normalization Perturbation (NP), perturbs the channel statistics of source domain low-level features to synthesize various latent styles, so that the trained deep model can perceive diverse potential domains and generalizes well even without observations of target domain data in training. This approach is motivated by the observation that feature channel statistics of the target domain images deviate around the source domain statistics. We further explore the style-sensitive channels for effective style synthesis. Normalization Perturbation only relies on a single source domain and is surprisingly simple and effective, contributing a practical solution by effectively adapting or generalizing classification DG methods to robust object detection. Extensive experiments demonstrate the effectiveness of our method for generalizing object detectors under real-world domain shifts.

## 1 Introduction

Object detection, a fundamental computer vision task, plays an important role in various safety-critical applications, including autonomous driving (Grigorescu et al., 2020), video surveillance (Raghunandan et al., 2018), and healthcare (Dusenberry et al., 2020). Deep learning has made great progress on in-domain data (Ren et al., 2015; Bochkovskiy et al., 2020; Fan et al., 2020; 2022) for object detection, but its performance usually degrades under domain shifts (Sakaridis et al., 2018; Michaelis et al., 2019), where the testing (target) data differ from the training (source) data.

Real-world domain shifts are usually brought by environment changes, such as different weather and time conditions, attributed by diverse contrast, brightness, texture, etc. Trained models usually overfit to the source domain style and generalize poorly in other domains, posing serious problems in challenging real-world usage such as autonomous driving. Figure 1(b) shows a large gap of feature channel statistics between two distinct domains, Cityscapes (Cordts et al., 2016) and Foggy Cityscapes (Sakaridis et al., 2018), especially in shallow CNN layers which preserve more style information (Zhou et al., 2020b; Pan et al., 2018). Deep models trained on the source domain cannot generalize well on the target domain, due to the discrepancy in feature channel statistics caused by the domain style overfitting.

Domain generalization (DG) (Muandet et al., 2013; Ghifary et al., 2016; Mahajan et al., 2021; Li et al., 2020) aims to solve this hard and significant problem. Major undertaking has been done

---

This work was done when Qi was the visiting scholar at MPII.
This research was supported by the Research Grant Council of the HKSAR under grant No. 16201420.

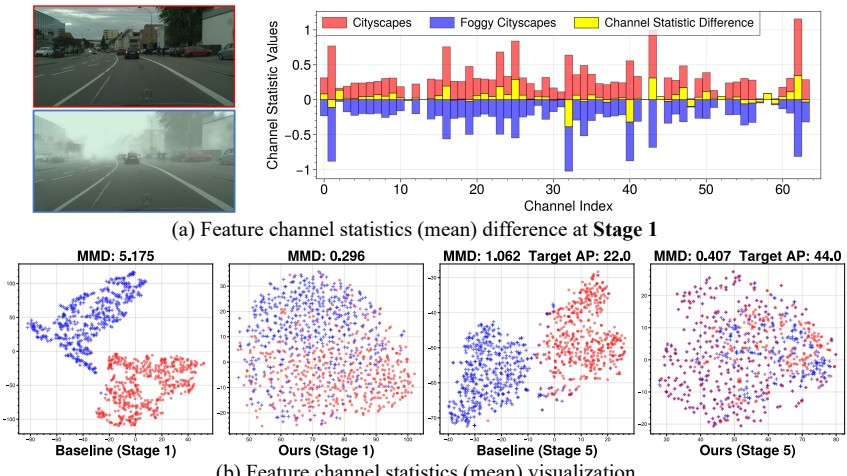

(a) Feature channel statistics (mean) difference at **Stage 1**

(b) Feature channel statistics (mean) visualization

Figure 1: Visualizations for feature channel statistics on Cityscapes (source domain, red) and Foggy Cityscapes (target domain, blue). (a) For two domain images with the same content but different styles, we show their feature channel statistics and differences on the pretrained backbone at stage 1 ("stage" denotes the backbone block). The statistics of the Foggy Cityscapes image are negated for better visualization. The feature channel statistics of the target domain image deviate around the source domain statistics. (b) The t-SNE (Van der Maaten & Hinton, 2008) map for the feature channel statistics. The model is trained on the source domain and evaluated on both domains. The distance between two domains is computed by Maximum Mean Discrepancy (Borgwardt et al., 2006) (MMD). After equipping Normalization Perturbation in shallow CNN layers, our model can effectively blend distinct domain style distributions. Thus our model generalizes much better on the target domain.

on improving domain generalization of classification models, where multiple source domains with large inter-image style variance are available for model training. However, less attention has been paid on robust object detection (Wang et al., 2021) which is of equal importance if not more in many visual perceptual systems. The closely related unsupervised domain adaptation (UDA) object detection (Schneider et al., 2020; Nado et al., 2020) has been widely studied, but it requires target domain images for model training, which is often infeasible for some online object detection systems.

Synthesizing new domains have been demonstrated as an effective solution for domain generalization in the classification task (Nuriel et al., 2021; Zhou et al., 2020b). The rationale behind is that the model can learn domain-invariant representations and generalizes well by perceiving a large variety of synthesized domains during training. But existing domain synthesis methods are all specifically designed for image classification, and it is non-trivial to directly apply these methods for robust object detection because of the task gap between classification and detection.

Specifically, the feature-level synthesis approach (Zhou et al., 2020b; Li et al., 2022) is effective for classification DG problem, but it requires multiple source domains with large style variance. However, there is usually only one single source domain for robust object detection due to the expensive annotation cost, which means that only relatively small style variance can exist. In this situation, previous feature-level synthesis approach cannot synthesize sufficient diverse domains. The image generation based synthesis approach (Jackson et al., 2019; Geirhos et al., 2018) can effectively address the single-source domain problem by leveraging an extra large-scale style image dataset (Kaggle) to synthesize diverse domains. But the image generation procedure may suffer from the potential destruction on image contents, which are essential for the hierachical object detection where large context diversity may be present. In this paper, we perform in-depth problem analysis for the under-explored robust object detection, and propose a novel domain style synthesis approach.

Figure 1(a) shows our motivation. Feature channel statistics of the target domain image deviate around the source domain statistics. Thus by perturbing the feature channel statistics of source domain images in the shallow CNN layers, we can effectively synthesize new domains. The perturbed feature statistics correspond to various latent domain styles, so that the trained model perceives diverse potential domains accordingly. Such perturbation enables deep models to learn domain-invariant representations where distinct domains can be effectively blended together in the learned feature

space. To further boost the performance, we have also explored the style-sensitive channels for effective domain style synthesis. Figure 1(b)[1] shows the distinct domains can be effectively blended by the perturbed channel statistics in shallow CNN layers. The learned deep CNN representations are thus more robust to the variations of different domain styles and generalize well on the target domain. Note that the model is only trained using a single source domain data without any access to the target domain style or data.

Our method normalizes features into different scales, thus called Normalization Perturbation (NP). NP perturbs the feature channel statistics of source domain images to effectively blend distinct domain style distributions, which is not limited by the small source domain style variance inherent in while does not affect the input images. Thus our model generalizes much better on robust object detection, which has small source domain style variance and large context diversity. Our NP is simple and effective, generalizes well under various real-world domain shifts, and outperforms previous DG and UDA methods on multiple dense prediction tasks. Our object detection DG method design contributes a practical solution by effectively adapting or generalizing existing feature perturbation based classification DG methods to robust object detection. In summary, our contributions are:

- We investigate the real-world domain shifts for robust object detection and highlight short-comings of existing domain synthesis approach. We contribute to researchers on adapting existing classification DG methods to effectively address the object detection DG problem.
- We propose to perturb the channel statistics of source domain features to synthesize various new domain styles, which enables a model to learn domain-variant representations for good domain generalization.
- Our method generalizes surprisingly well under various real-world domain shifts and is very simple, without any extra input, learnable parameters or loss. Extensive experiments are conducted to verify the effectiveness of our method.

## 2 RELATED WORKS

Domain Generalization (DG) (Hendrycks & Dietterich, 2018; Venkateswara et al., 2017), which targets at generalizing models to unseen domains, relies on source data typically consisting of multiple distinct domains. DG has been mainly studied in the context of object recognition task (Du et al., 2020a;b; Yue et al., 2019). DG of dense prediction tasks (Seemakurthy et al., 2022) (Hasan et al., 2021; Lin et al., 2021) has attracted increasing interests because of its wide real-world applications. The closely related unsupervised domain adaptation (UDA) (Long et al., 2016; Damodaran et al., 2018) has been widely studied on dense prediction tasks (VS et al., 2021; Kim et al., 2019) for real-world applications, which aims to generalize model to the target domain by accessing its unlabeled images. Both DG and UDA share significant overlap in technicalities, such as domain alignment (Jia et al., 2020; Zhou et al., 2021), self-supervised learning (Xie et al., 2020; Iqbal & Ali, 2020), feature disentanglement (Wang et al., 2020b; Yang et al., 2019), and data augmentation (Choi et al., 2019; Wang et al., 2022). Our method closely relates to the following works.

**Normalization-based Methods.** The normalization layers are leveraged to improve model generalization ability. Various normalization variants have been proposed, such as domain-specific BatchNorm (Liu et al., 2020; Segu et al., 2020), AdaBN (Li et al., 2016), PreciseBN (Wu & Johnson, 2021), Instance-Batch Normalization (Choi et al., 2021; Pan et al., 2018), Adversarially Adaptive Normalization (Fan et al., 2021), Switchable Normalization (Luo et al., 2019), and Semantic Aware Normalization (Peng et al., 2022). The test-time adaptation (Wang et al., 2020a; Zhang et al., 2021) attempts to estimate accurate normalization statistics for the target domain during testing. These methods fit normalization layers to the specific target domains, while our method normalizes features into different scales to implicitly synthesize arbitrary new domains, and is optimization-free.

**Synthesizing New Domains.** Data augmentation (Volpi & Murino, 2019; Otálora et al., 2019) has been widely used to synthesize new domains in DG and UDA. Some methods synthesize new domain images using image-to-image translation models, such as the random (Xu et al., 2020) or learnable augmentation networks (Carlucci et al., 2019; Zhou et al., 2020a), and style transfer models (Yue et al., 2019; Somavarapu et al., 2020). Other works propose to perform implicit domain synthesis through the feature-level augmentation (Zhou et al., 2020b; Jin et al., 2021; Mancini et al., 2020;

---

[1]For all t-SNE visualizations in this paper, the features from multiple models are mapped jointly into a unified space but are separately visualized for clarity.

Table 1: Difference between our methods and closely related classification DG methods, where all methods are represented by $y = x + \varepsilon_x$. The $\mu_\star$ and $\sigma_\star$ denote the feature channel mean and standard deviation, respectively. The $\lambda$ and $(\alpha, \beta)$ denote the Beta and Gaussian distribution, respectively.

| Method | $\varepsilon_x$ | $\sigma^2(\varepsilon_x)$ |
|---|---|---|
| Mixstyle | $\frac{\lambda(x-\mu_1)}{\sigma_1}(\sigma_2 - \sigma_1) + \lambda(\mu_2 - \mu_1)$ | $[\frac{(x-\mu_1)}{\sigma_1}(\sigma_2 - \sigma_1) + (\mu_2 - \mu_1)]^2 \sigma_\lambda^2$ |
| DSU | $\frac{\alpha(x-\mu_1)}{\sigma_1}\sum_\sigma(x) + \beta\sum_\mu(x)$ | $[\frac{(x-\mu_1)}{\sigma_1}\sum_\sigma(x)]^2\sigma_\alpha^2 + [\sum_\mu(x)]^2\sigma_\beta^2$ |
| NP (Ours) | $(\alpha - 1)x + (\beta - \alpha)\mu_1$ | $x^2\sigma_\alpha^2 + (\sigma_\alpha^2 + \sigma_\beta^2)\mu_1^2$ |
| NP+ (Ours) | $(\alpha - 1)x + [\sum_\mu(x)](\beta - \alpha)\mu_1$ | $x^2\sigma_\alpha^2 + [\sum_\mu(x)]^2(\sigma_\alpha^2 + \sigma_\beta^2)\mu_1^2$ |

Tang et al., 2021; Nuriel et al., 2021; Gong et al., 2019) to mix CNN feature statistics of distinct domains to significantly improve the domain synthesis efficiency.

The above methods rely on image generation or multiple source domains for new domain synthesis, and thus the efficiency or effectiveness is limited. On the other hand, our method only relies on a single source domain to diversify by perturbing the feature channel statistics to produce various latent domain styles.

While SFA (Li et al., 2021) performs the activation-wise feature perturbation, which may destroy meaningful image contents, our method performs the channel-wise feature statistics perturbation and keeps the image contents unchanged. Note that some methods also perform feature statistics perturbation, *e.g.*, Mixstyle (Zhou et al., 2020b), DSU (Li et al., 2022), and pAdaIN (Nuriel et al., 2021). But they all rely on multiple source domains with large style variance to synthesize diverse domains. Their effectiveness is unknown for object detection under real-world domain shifts. In contrast, our method is surprisingly effective for the real-world applications of robust object detection.

## 3 PROBLEM ANALYSIS

We aim at out-of-domain generalization for robust object detection. Object detectors are trained on source domains and tested on different target domains unknown during training. In this section, we analyze the shortcomings of existing domain synthesis approach on robust object detection to motivate our method.

Robust object detection is much harder than the classification domain generalization problem. The latter usually contains multiple source domains with large style variance for training, and the pertinent training images are usually dominated by one target object in a relatively simple background. By contrast, with small domain style variance and large context diversity, robust object detection usually has only one single source domain for training due to the expensive annotation cost, where the training images usually contain multiple objects in various scales in front of complex background. Thus, existing classification DG methods cannot handle well the robust object detection problem, especially for the domain synthesis based methods.

We conduct empirical studies to demonstrate the shortcomings of existing domain synthesis methods on robust object detection, so as to motivate our Normalization Perturbation method on the robust object detection task. The model is trained on the source domain and evaluated on the unseen target domain, *i.e.*, Cityscapes (C) (Cordts et al., 2016) → Foggy Cityscapes (F) (Sakaridis et al., 2018), and Sim10k (S) (Johnson-Roberson et al., 2017) → Cityscapes (C). We use the Faster R-CNN (Ren et al., 2015) model (with ImageNet (Deng et al., 2009)-pretrained ResNet-50 (He et al., 2016) backbone) as the baseline. The detection performance is evaluated using the mean average precision (mAP) metric with the threshold of 0.5. For better comparisons, we also conduct classification DG experiments following the experiment setups in Mixstyle (Zhou et al., 2020b) and DSU (Li et al., 2022), which are closely related methods to our NP method. Refer to the appendix for full experimental details.

**Small Domain Style Variance Restricts Feature-level Domain Synthesis.** To facilitate the method illustration, we rewrite the feature-level domain synthesis methods formulation from $y = \sigma_{new}\frac{x-\mu}{\sigma} + \mu_{new}$ ($x$ is the original feature with its channel mean $\mu$ and standard deviation $\sigma$, and $y$ is the output feature with the synthesized new channel mean $\mu_{new}$ and standard deviation $\sigma_{new}$.) to

$$y = x + [(\frac{\sigma_{new}}{\sigma} - 1)x - \frac{\sigma_{new}}{\sigma}\mu + \mu_{new}] = x + \varepsilon_x$$

Table 2: Comparison results on the object detection and classification domain generalization tasks. The $[\sum_\mu(x)]^2$ and $[\sum_\sigma(x)]^2$ respectively denote the variance of the feature statistics (mean and standard deviation) of all images in the entire dataset train set.

| Source | $[\sum_\mu(x)]^2$ | $[\sum_\sigma(x)]^2$ | Target | Baseline | Mixstyle | DSU | CycConf |
|---|---|---|---|---|---|---|---|
| Cityscapes | 1.657 | 1.870 | Foggy Cty | 22.0 | 30.1 | 34.1 | 40.9 |
| Sim10k | 16.507 | 4.309 | Cityscapes | 32.8 | 46.4 | 49.3 | 52.1 |
| PACS Train | 37.163 | 8.786 | PACS Test | 79.5 | 83.7 | 84.1 | - |

This equation can be represented as $y = x + \varepsilon_x$, where $\varepsilon_x = (\frac{\sigma_{new}}{\sigma} - 1)x - \frac{\sigma_{new}}{\sigma}\mu + \mu_{new}$ denotes the perturbation (random noise) given the original input feature $x$. The perturbation term $\varepsilon_x$ depends on the given $x$ and how methods perturb the feature $x$.

We use Mixstyle and DSU as the example to illustrate these feature statistic perturbation methods are suboptimal for robust object detection, as shown in Table 1. When $(\sum_\sigma(x) \to 0, \sum_\mu(x) \to 0)$, both Mixstyle and DSU have $(\sigma^2(\varepsilon_x) \to 0, \mu(\varepsilon_x) \to 0)$, and thus fail to perturb the feature $x$ where the method formulation $y = x + \varepsilon_x$ collapses to $y = x$. Mixstyle and DSU rely on large variance in inter-image feature statistics $(\sum_\sigma(x), \sum_\mu(x))$ to produce sufficient feature noise $\sigma^2(\varepsilon_x)$. Thus it works well on the classification DG datasets, e.g., PACS, which contains multiple source domains with large inter-image style variance. Unfortunately, this assumption cannot always hold. In robust object detection, there is usually only one source domain due to the expensive annotation cost, which means that only relatively small inter-image style variance can be present. In this situation, Mixstyle and DSU cannot produce sufficient noises to perturb the feature map.

Table 2 shows the feature statistic variance of different datasets, i.e., Cityscapes, Sim10k and PACS (Li et al., 2017). Both Cityscapes and Sim10k[2] datasets have much smaller feature statistic variance than PACS. DSU achieves SOTA on the classification DG benchmark PACS with Mixstyle performing comparably. This is because the large style variance of multiple source domains enables these methods to produce decent feature noise for sufficient domain style synthesis. By contrast, DSU and Mixstyle perform much worse than the robust object detection SOTA method CycConf (Wang et al., 2021). Here, smaller feature statistic variance results in larger performance gap between DSU/Mixstyle and the SOTA CycConf method. This performance gap indicates that DSU and Mixstyle do not have sufficient diversity in style synthesis when the inter-image style variance is too small.

**Large Context Diversity Restricts Image-level Domain Synthesis.** The image-level domain synthesis methods (Jackson et al., 2019; Geirhos et al., 2018) leverage large-scale style image dataset to generate diverse stylized training images through style transfer techniques (Huang & Belongie, 2017). Unfortunately, such image generation procedure may destroy the content structures underlying the original images. Figure 4 (second column) shows significant lost of meaningful object details in the stylized training images under large context diversity. The trained model substantially sacrifices the source domain performance where its generalization ability improvement is also limited.

The above two analyses show that existing classification DG methods cannot be directly applied for robust object detection, especially for the domain synthesis approach, which motivate us to propose a novel domain synthesis method to address the two issues, namely, small domain style variance and large context diversity, in robust object detection.

## 4 METHOD AND ANALYSIS

It has been known that feature channel statistics, e.g., mean and standard deviation, are closely related to image styles, where changing feature channel statistics can be regarded as implicitly changing the input image styles. The Adaptive Instance Normalization (AdaIN) (Huang & Belongie, 2017) achieves arbitrary style transfer through the feature channel statistics normalization and transformation. Given a mini-batch $B$ of CNN features $x \in \mathcal{R}^{B \times C \times H \times W}$ with $C$ channels and $H \times W$ spatial size from

---

[2]Although Sim10k dataset is a single domain dataset, it contains diverse styles synthesized by graphics engine, e.g., daytime, night, dawn, dusk, clear, snowy and rainy. Thus Sim10k has larger inter-image feature statistics variance than Cityscapes, but is still smaller than that of PACS.

$$\text{IN} \rightarrow \text{Stage 1} \xrightarrow{\boldsymbol{x_1}} \boldsymbol{\alpha_1 x_1 + (\beta_1 - \alpha_1)\mu_{1,c}} \rightarrow \text{Stage 2} \xrightarrow{\boldsymbol{x_2}} \boldsymbol{\alpha_2 x_2 + (\beta_2 - \alpha_2)\mu_{2,c}} \rightarrow \text{Stage 3-5} \rightarrow \text{OUT}$$

$$x_i \in \mathcal{R}^{B \times C \times H_i \times W_i} \qquad \mu_{i,c} = \frac{1}{H_i W_i}\Sigma_{H_i}\Sigma_{W_i}x_i \in \mathcal{R}^{B \times C} \qquad \alpha_i, \beta_i \sim Gaussian(1, 0.75) \in \mathcal{R}^{B \times C}$$

Figure 2: Our Normalization Perturbation (NP) is applied at shallow CNN layers only during training. NP is enabled with probability $p = 0.5$.

the content images, AdaIN can be formulated as:

$$y = \sigma_s \frac{x - \mu_c}{\sigma_c} + \mu_s, \tag{1}$$

where both $\{\mu_c, \sigma_c\} \in \mathcal{R}^{B \times C}$ and $\{\mu_s, \sigma_s\} \in \mathcal{R}^{B \times C}$ are feature channel statistics, estimated from the input content images and style images, respectively. The normalized features $y$ can be decoded into the stylized content images. AdaIN provides a feasible and efficient way to implicitly change image styles in the feature space.

## 4.1 NORMALIZATION PERTURBATION METHOD

Our proposed method, Normalization Perturbation (NP), perturbs the feature channel statistics of training images by inserting random noises. Formally, NP can be formulated as:

$$y = \sigma_s^{\star}\frac{x - \mu_c}{\sigma_c} + \mu_s^{\star}, \qquad \sigma_s^{\star} = \alpha\sigma_c, \qquad \mu_s^{\star} = \beta\mu_c \tag{2}$$

where $\{\mu_c, \sigma_c\} \in \mathcal{R}^{B \times C}$ are the channel statistics, mean and standard deviation, estimated on the input features. The $\{\alpha, \beta\} \in \mathcal{R}^{B \times C}$ are random noises drawn from the Gaussian distribution. This equation can be further simplified as:

$$y = \alpha x + (\beta - \alpha)\mu_c. \tag{3}$$

As shown in Figure 2, NP is applied at shallow CNN layers (following the backbone stage 1 and stage 2). NP is enabled with probability $p$ only in the training stage.

Our NP method is fundamentally different from conventional normalization methods (Huang & Belongie, 2017; Ioffe & Szegedy, 2015; Ulyanov et al., 2016; Wu & He, 2018), whose affine parameters $\{\mu_s, \sigma_s\}$ are learned from the training set or estimated from extra input style images. While NP affine parameters $\{\mu_s^{\star}, \sigma_s^{\star}\}$ are obtained by perturbing the input feature channel statistics, they are obtained without relying on extra style inputs and are optimization-free. The perturbed affine parameters can be regarded as the channel statistics corresponding to diverse latent domain styles, enabling models to learn domain-invariant representations and preventing them from style overfitting.

In NP, all channel statistics are randomly perturbed with the same noise distribution. We further propose Normalization Perturbation Plus (NP+) to adaptively control the noise magnitude in different channels, based on the feature statistic variance across different images. Such adaptive perturbation is motivated by the observation that some channels significantly vary as the domain style changes. We thus apply more noise on these style-sensitive channels. Specifically, we use the mini-batch of $B$ feature channel statistics $\mu_c = \{\mu_c^1, ..., \mu_c^b, ..., \mu_c^B\}$ to compute the statistic variance $\Delta \in \mathcal{R}^{1 \times C}$: $\Delta = \frac{1}{B}\sum_{b=1}^{B}(\mu_c^b - \bar{\mu}_c)^2$, where $\bar{\mu}_c = \frac{1}{B}\sum_{b=1}^{B}(\mu_c^b)$. Then we use the normalized statistic variance $\delta = \Delta/\texttt{max}(\Delta) \in \mathcal{R}^{1 \times C}$ to control the injected noise magnitude for each channel:

$$y = \alpha x + \delta(\beta - \alpha)\mu_c, \tag{4}$$

where $\texttt{max}$ is the maximum operation. When applying NP+, we use the photometric data augmentation (Color Jittering, GrayScale, Gaussian Blur and Solarize, only for NP+ by default) to generate pseudo domain styles to facilitate the exploration on style-sensitive channels.

## 4.2 ADDRESSING ROBUST OBJECT DETECTION

Our Normalization Perturbation can be implemented as a plug-and-play component in modern CNN models to effectively solve the domain style overfitting problem, where the domain style has low variance while large context diversity exists typical of robust object detection.

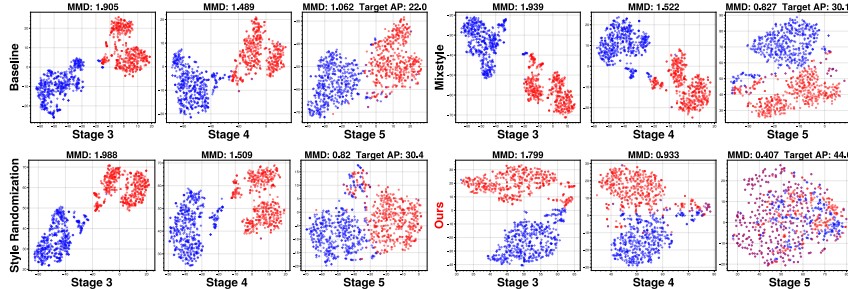

Figure 3: The t-SNE visualization for the feature channel statistics of different methods on Cityscapes (source domain) and Foggy Cityscapes (target domain). The target domain performance is presented.

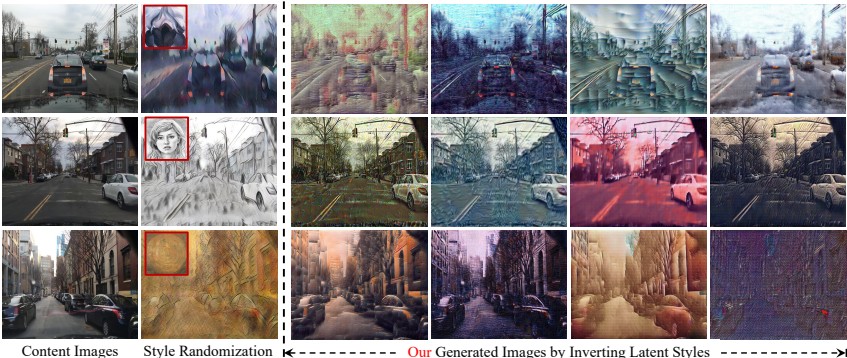

Figure 4: Latent style visualization. The second column represents stylized images generated by style randomization. The 3-6 columns show that the perturbed channel statistics of the training image features are inverted back to the image space.

**Effective Domain Blending.** Our NP can effectively blend feature channel statistics of different domains, corresponding to learning better domain-invariant representations. Figure 3 shows that the Ours model trained with NP can effectively reduce the learned distribution distance between source and target domains, especially on deep CNN layers. Compared to other methods, NP results in smaller cross-domain distribution distance and better generalization performance on target domains.

**Diverse Latent Styles.** Our NP can effectively diversify latent styles. Table 1 shows NP's $\sigma^2(\varepsilon_x)$ does not rely on inter-image statistic variance, and thus it provides more diversified style diversity during synthesis even under the small domain style variance typical of robust object detection. On the other hand, although our NP+ method is also related to inter-image statistic variance, its $\sigma^2(\varepsilon_x) = x^2\sigma_\alpha^2 + [\sum_\mu(x)]^2(\sigma_\alpha^2 + \sigma_\beta^2)\mu_1^2$ has the $x^2\sigma_\alpha^2$ term to ensure its scale is sufficiently large. Our NP+ only partially uses inter-image style variance while achieving the best performance. This indicates that proper usage of inter-image style variance alongside our NP method is important for DG tasks, especially for robust object detection.

Sufficiently large perturbation variance $\sigma^2(\varepsilon_x)$ is the key to synthesizing higher style diversity for achieving better domain generalization ability. Table 4 shows the ablation study on noise hyperparameters of our NP method, reflecting different noise variance $\sigma^2(\varepsilon_x)$ scale. Our NP is also insensitive to the noise types and hyperparameters, performing well with Beta, Uniform and Gaussian noises with different $\alpha$ and $\beta$.

For better understanding, we map the NP perturbed feature channel statistics back into the image space using the feature inverting technique (Zeiler & Fergus, 2014). Figure 4 shows that the generated latent styles effectively diversify the style scopes, covering various potential unseen domain styles in real-world environments, *e.g.*, dawn, dusk, night times and foggy, rainy, snowy weathers.

Figure 4 also shows the stylized images of the image-level synthesis method (Geirhos et al., 2018), which transfers diverse styles into content images for model training. But the generated stylized images fail to preserve the content structure details inherent in the original images. In contrast, our NP generated latent styles can often cover the potential synthesized domain styles generated by style randomization, while simultaneously maintaining high content fidelity to original images.

Table 3: Experimental studies on Normalization Perturbation (NP). SR denotes style randomization. Models are trained on C (Cityscapes), and evaluated on C, F (Foggy Cityscapes) and B (BDD100k).

| Method | C | F | B |
|---|---|---|---|
| Baseline | 58.0 | 22.0 | 21.8 |
| Image SR | 51.9 | 30.4 | 26.0 |
| Feat SR | 58.2 | 42.0 | 29.0 |
| NP | **58.7** | 44.0 | 30.1 |
| NP+ | 58.3 | **46.3** | **32.8** |

(a) Synthesis method effect.

| Method | C | F | B | C | F | B | C | F | B |
|---|---|---|---|---|---|---|---|---|---|
| | without NP & DA | | | with DA | | | **with NP** | | |
| pAdaIN | 58.5 | 27.6 | 25.1 | 58.1 | 37.5 | 30.6 | 58.2 | 44.5 | 30.4 |
| Mixstyle | 57.7 | 30.1 | 26.5 | 57.5 | 40.2 | 30.8 | 58.3 | 44.7 | 30.6 |
| DSU | 58.5 | 34.1 | 27.2 | 58.2 | 43.1 | **31.4** | 58.4 | **44.9** | 31.0 |
| Baseline | 58.0 | 22.0 | 21.8 | 57.2 | 35.5 | 30.5 | **58.7** | 44.0 | 30.1 |

(b) **Classification DG methods benefit more from NP than DA.**

Table 4: Ablation studies on noise hyperparameters of our NP method.

| $\alpha$ | B(0.75, 0.75) | U(0, 2.0) | G(1, 0.1) | G(1, 0.25) | G(1, 0.5) | G(1, 0.75) | G(1, 1) | G(1, 0.75) | G(1, 0.1) |
|---|---|---|---|---|---|---|---|---|---|
| $\beta$ | B(0.75, 0.75) | U(0, 2.0) | G(1, 0.1) | G(1, 0.25) | G(1, 0.5) | G(1, 0.75) | G(1, 1) | G(1, 0.1) | G(1, 0.75) |
| C | **59.0** | 58.4 | **59.0** | 58.5 | 58.3 | 58.7 | 57.4 | 58.3 | 58.2 |
| F | 43.0 | 42.0 | 29.5 | 38.5 | 40.1 | 44.0 | **44.3** | 34.9 | 39.6 |
| B | 29.5 | 28.9 | 26.6 | 28.2 | 29.6 | 30.1 | **30.2** | 29.0 | 26.8 |

**High Content Fidelity.** Our NP processes feature channel statistics while faithfully preserving image and feature spatial structures. Note that image-level domain synthesis methods may destroy the content structures of the original images in the image generation procedure. Besides, NP trains deep models with numerous content-style combinations in the high-dimensional feature space, which is much more efficient and effective than the image-level methods, whose styles are deterministic, limited and that their style augmentation is only performed on the low-dimensional image space. Table 3(a) shows the comparisons between image- and feature-level domain synthesis methods. Image-level style randomization (Geirhos et al., 2018) sacrifices the source domain performance due to its potential damage on image contents, although this method effectively improves the detection performance on unseen target domains. Our NP method can be regarded as a variant of AdaIN, which has been widely shown (Huang & Belongie, 2017; An et al., 2021) being able to preserve well the image content information by keeping the normalized feature maps unchanged.

**Benefiting Other Methods.** Our NP can be regarded as a better generalization of existing SOTA methods. Table 3(b) shows that NP can boost the performance of other classification DG methods on robust object detection, by perturbing the feature statistics to synthesize sufficient diverse domain styles. Specifically, when combining our NP method with other DG methods, the source domain performance on Cityscapes (C) is almost unchanged while always better than the performance of baseline. The generalization performance on target domains Foggy Cityscapes/BDD100k (F/B) is significantly boosted with 16.9↑/5.3↑, 14.6↑/4.1↑, 10.8↑/3.8↑ AP50 improvements respectively for pAdaIN, Mixstyle and DSU, achieving better performance than the baseline model with NP method.

**Discussion on Data Augmentation.** The image-level data augmentation can be regarded as a special case of feature-level augmentation, if we treat the input image as a feature map with the low-dimensional RGB color channels. Thus such data augmentation is less effective than our feature-level perturbation, which is performed in the high-dimensional feature channels. Table 3(b) shows that the data augmentation (DA) significantly improves the generalization performance on different target domains. Our NP method boosts more performance improvement than data augmentation in most settings. Our NP method also benefits from data augmentation and the resulting NP+ method achieves the best generalization performance of 46.3/32.8 AP50 on Foggy Cityscapes/BDD100k.

## 5 COMPARISON EXPERIMENTS

We compare our method to object detection DG/UDA methods. The experiment setting details are also included in the appendix.

### 5.1 ROBUST OBJECT DETECTION

We follow CycConf (Wang et al., 2021) to train and evaluate models on the robust object detection benchmark. Specifically, there are two settings: Domain Shift by Time of Day, where the model is trained on BDD100k (Yu et al., 2020) daytime/night train set and evaluated on the night/daytime val

Table 5: Robust object detection results.

| | BDD Day → Night | | | BDD Night → Day | | | WaymoL → BDD | | | WaymoR → BDD | | |
| --- | --- | --- | --- | --- | --- | --- | --- | --- | --- | --- | --- | --- |
| | AP | AP50 | AP75 | AP | AP50 | AP75 | AP | AP50 | AP75 | AP | AP50 | AP75 |
| Faster R-CNN | 17.84 | 31.35 | 17.68 | 19.14 | 33.04 | 19.16 | 10.07 | 19.62 | 9.05 | 8.65 | 17.26 | 7.49 |
| + CycConsist | 18.35 | 32.44 | 18.07 | 18.89 | 33.50 | 18.31 | 11.55 | 23.44 | 10.00 | 9.11 | 17.92 | 7.98 |
| + CycConf | 19.09 | 33.58 | 19.14 | 19.57 | 34.34 | **19.26** | 12.27 | 26.01 | 10.24 | 9.99 | 20.58 | 8.30 |
| + NP (Ours) | 20.73 | 36.22 | 20.85 | 19.32 | 34.42 | 18.63 | 17.85 | 35.34 | 15.52 | 14.97 | 29.42 | 13.11 |
| + NP+ (Ours) | **20.97** | **36.76** | **21.10** | **19.73** | **35.30** | 19.19 | **21.18** | **42.16** | **18.67** | **19.64** | **38.69** | **17.07** |

set, and Cross-Camera Domain Shift, where the model is trained on Waymo (Sun et al., 2020) Front Left/Right train set and evaluated on BDD100k night val set. Table 5 shows our NP outperforms previous SOTA CycConf (Wang et al., 2021) on all domain shift settings, especially on the Waymo Front Left/Right to BDD100k Night setting, where our NP improves the SOTA performance from 12.27/9.99 to 17.85/14.97 AP. Our NP+ further boosts the performance to a new SOTA. Note that our method is general without any assumption on the inputs, while CycConf is designed to operate on videos, requiring consecutive video frame pairs to form a time cycle.

## 5.2 UNSUPERVISED DOMAIN ADAPTIVE DETECTION

Unsupervised domain adaptive object detection models are trained on labeled source domain and unlabeled target domain. We consider two popular adaptation settings: Sim10k (Johnson-Roberson et al., 2017) to Cityscapes (S → C) and Cityscapes (Cordts et al., 2016) to Foggy Cityscapes (Sakaridis et al., 2018) (C → F) adaptations. Table 6 shows that our method significantly outperforms the UDA methods by a large margin **even without accessing the target domain data**, where the UDA methods (DA-Faster (Chen et al., 2018),SC-DA (Zhu et al., 2019),ViSGA (Rezaeianaran et al., 2021), EPM (Hsu et al., 2020), and CycConf (Wang et al., 2021) with self-supervised rotation) all require unlabeled target domain images for model training. We also apply closely related classification DG methods (SFA (Li et al., 2021), pAdaIN (Nuriel et al., 2021), Mixstyle (Zhou et al., 2020b), and DSU (Li et al., 2022)) on our baseline for a fair comparison. Our method substantially outperforms other DG methods on UDA object detection, especially on the C → F setting, where Cityscapes (C) source domain has smaller

Table 6: UDA object detection AP50 performance on ResNet-50 backbone except [†] which are ResNet-101.

| Method | Target | S → C | C → F |
| --- | --- | --- | --- |
| FR-CNN | ✗ | 31.9 | 22.8 |
| DA-Faster | ✓ | 41.9 | 32.0 |
| SC-DA | ✓ | 45.1 | 35.9 |
| ViSGA | ✓ | 49.3 | 43.3 |
| EPM[†] | ✓ | 51.2 | 40.2 |
| CycConf[†] | ✓ | 52.4 | 41.5 |
| Our Baseline | ✗ | 32.8 | 22.0 |
| BIN | ✗ | 44.3 | 28.4 |
| IBN | ✗ | 47.4 | 31.2 |
| SFA | ✗ | 38.4 | 25.3 |
| pAdaIN | ✗ | 43.7 | 27.6 |
| Mixstyle | ✗ | 46.4 | 30.1 |
| DSU | ✗ | 49.3 | 34.1 |
| NP (Ours) | ✗ | 54.1 | 44.0 |
| NP+ (Ours) | ✗ | 58.7 | 46.3 |

domain style variance. Our method also substantially outperforms BIN (Nam & Kim, 2018) and IBN (Pan et al., 2018), whose normalization layer integration manner is fundamentally different from our normalization perturbation manner.

## 6 CONCLUSION

SOTA classification DG methods perform unsatisfactorily on robust object detection, since they are limited by the small domain style variance and large context diversity in the object detection DG task. We propose Normalization Perturbation (NP) to perturb the channel statistics of source domain features to synthesize various latent styles. NP is not limited by the small source domain style variance in the input images and does not destroy image structures. The trained deep model can perceive diverse potential domains and thus generalizes well on unseen domains thanks to the learned domain-invariant representations. Our NP method only relies on a single source domain to generalize on diverse real-world domains, which is effective and easy to implement, while boosting the performance of other classification DG methods on robust object detection. Extensive analysis and experiments verify the effectiveness of our Normalization Perturbation.

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

## A    EXPERIMENT SETTING DETAILS

Given the simplicity, anyone can implement NP/NP+ with ease. Our experiments are all based on public codebases. While we will make our codes and data public upon acceptance, a competent practitioner can easily reproduce our results. The following gives all the necessary details.

In our main paper, all object detection experiments (Table 2–6 of the main paper) are based on the CycConf (Wang et al., 2021) public codebase (Confusion) and mmdetection (Chen et al., 2019) public codebase (MMDetection) (UDA object detection strong baseline). We use their default settings in our experiments, except that we adjust the training iterations for some datasets to avoid overfitting. Table 7 shows the object detection experiment setting details.

## B    NORMALIZATION PERTURBATION IMPLEMENTATION

As shown in the below code block, our Normalization Perturbation (NP) and Normalization Perturbation Plus (NP+) can be easily inserted into popular CNN backbones.

```python
import torch
import random

# ResNet backbone equipped with our Normalization_Perturbation.
Class ResNet_backbone():

    def __init__(self, ...):
        ...

    def forward(x):
        p = random.random()

        x = self.stage_1(x)
        if p < 0.5 and self.training:
            # Can be replaced by Normalization_Perturbation_Plus
            x = Normalization_Perturbation(x)
```

Table 7: Object detection experiment setting details.

| Config | Table 5 | Table 2,3,4,6 |
|---|---|---|
| description | Default object detection setting | |
| codebase | Detectron2-CycConf Confusion | |
| pretraining data | ImageNet | |
| backbone | ResNet-50 | |
| backbone norm | FrozenBN | |
| backbone freeze_at | 2 | |
| detector | Faster R-CNN with FPN | |
| batch size | 16 | |
| # GPU | 4 | |
| LR scheduler | WarmupMultiStepLR | |
| base LR | 0.02 | |
| gamma | 0.1 | |
| momentum | 0.9 | |
| weight decay | 0.0001 | |
| warmup method | linear | |
| warmup iters | 1000 | |
| warmup factor | 1.0 / 1000 | |
| LR steps | (36000, 48000) | (12000, 16000) |
| training iterations | 52500 | 17500 |
| training (min, max) size | (800, 1333) | |
| testing (min, max) size | (800, 1333) | |

```python
        x = self.stage_2(x)
        if p < 0.5 and self.training:
            # Can be replaced by Normalization_Perturbation_Plus
            x = Normalization_Perturbation(x)

        x = self.stage_3(x)
        x = self.stage_4(x)
        x = self.stage_5(x)

        return x

def Normalization_Perturbation(feat):
    # feat: input features of size (B, C, H, W)

    feat_mean = feat.mean((2, 3), keepdim=True) # size: B, C, 1, 1
    ones_mat = torch.ones_like(feat_mean)

    alpha = torch.normal(ones_mat, 0.75 * ones_mat) # size: B, C,
    ↪  1, 1
    beta = torch.normal(ones_mat, 0.75 * ones_mat) # size: B, C,
    ↪  1, 1

    output = alpha * feat - alpha * feat_mean + beta * feat_mean
    return output # size: B, C, H, W

def Normalization_Perturbation_Plus(feat):
    feat_mean = feat.mean((2, 3), keepdim=True)
    ones_mat = torch.ones_like(feat_mean)
    zeros_mat = torch.zeros_like(feat_mean)

    mean_diff = torch.std(feat_mean, 0, keepdim=True)
    mean_scale = mean_diff / mean_diff.max() * 1.5
```

```
    alpha = torch.normal(ones_mat, 0.75 * ones_mat)
    beta = 1 + torch.normal(zeros_mat, 0.75 * ones_mat) *
    ↪   mean_scale

    output = alpha * feat - alpha * feat_mean + beta * feat_mean
    return output
```

When applying NP+ in the backbone, we apply photometric data augmentation to generate pseudo domain images to facilitate the domain style synthesis by exploring style-insensitive channels. Specifically, we adopt the MoCoV3 (Chen* et al., 2021) data augmentation implementation in our method, as shown in the below code block.

```python
from PIL import ImageFilter, ImageOps
import random
import torchvision.transforms as transforms

photometric_data_augmentation = [
    transforms.RandomApply([
        transforms.ColorJitter(0.4, 0.4, 0.2, 0.1)
    ], p=0.8),
    transforms.RandomGrayscale(p=0.3),
    transforms.RandomApply([GaussianBlur([.1, 2.])], p=0.3),
    transforms.RandomApply([Solarize()], p=0.3),
]

class GaussianBlur(object):
    "Gaussian blur augmentation from SimCLR"

    def __init__(self, sigma=[.1, 2.]):
        self.sigma = sigma

    def __call__(self, x):
        sigma = random.uniform(self.sigma[0], self.sigma[1])
        x = x.filter(ImageFilter.GaussianBlur(radius=sigma))
        return x

class Solarize(object):
    "Solarize augmentation from BYOL"

    def __call__(self, x):
        return ImageOps.solarize(x)
```

