# OpenReview forum: "Towards Robust Object Detection Invariant to Real-World Domain Shifts"
_ICLR.cc/2023/Conference — ICLR 2023 poster_

### Official Review · Reviewer_kYbi · 2022-10-24

**Confidence:** 3
**Correctness:** 3
**Technical Novelty And Significance:** 3
**Empirical Novelty And Significance:** Not applicable
**Recommendation:** 8

**Clarity, Quality, Novelty And Reproducibility:**

Quality: Good.

Clarity: Good.

Novelty: Good.

Reproducibility: Good.

**Strength And Weaknesses:**

Strength:

(1) The proposed method is simple but effective, demonstrating its power in several tasks including robust object detection, UDA object detection, and DG image classification.

(2) The proposed method is easy to implement, and relies on only one source domain.

(3) The paper is well organized and clearly presented.

Weaknesses:

(1) For domain generalization, the combination of IN and BN layers has also been proved to be simple and effective, such as the IBN network and BIN method. It would be better to also discuss and compare such kind of methods.

(2) While traditional data augmentation methods can be considered as perturbations on image level, there lacks a discussion and comparison on why feature-level perturbation should be better than image-level augmentation to enlarge the diversity of the training data.

(3) It is not very convincing that object detection is hard to be trained with multiple source domains, as even in public domains there are already several datasets with different styles. While it is an advantage that the proposed method only relies on one single source domain for training, it would still be interesting to see experimental comparisons with all methods trained on multiple domains to see the difference.

(4) There are a few other generalizable object detection studies which are not reviewed in the paper, such as:

Lin et al. Domain-Invariant Disentangled Network for Generalizable Object Detection, ICCV 2021.

Hasan et al. Generalizable pedestrian detection: The elephant in the room, CVPR 2021.


**Summary Of The Paper:**

In this paper, a method called Normalization Perturbation (NP) is proposed for robust object detection under domain shift. The proposed method perturbs the channel statistics of source domain features to synthesize various latent styles. Extensive analysis
and experiments verify the effectiveness of the proposed NP method.


**Summary Of The Review:**

The paper proposes a simple but effective method for robust object detection. To me this work is novel and deserves further attention of the community.

---

> ### Author Response · Authors · 2022-11-18
> **Author Responses to Reviewer kYbi**
>
> ----
>
> **Q3. Evaluation on multiple training domains.**
>
> **A3.** Thanks for your insightful suggestion.
>
> First, when applying photometric data augmentation (i.e., Color Jittering, GrayScale, Gaussian Blur and Solarize) to training images, it explicitly changes the image styles and thus the model can be treated as trained on multiple domains.
> **The results above show that our method still outperforms other methods even trained with photometric data augmentation.**
>
>
> **Second, to further validate our method in the multiple training domains scenario, we use the Cityscapes, Foggy Cityscapes, KITTI and Sim10k datasets as the source domains.** Because the Sim10k and KITTI contain only the car class label, we train and evaluate the model only on the car class. We use BDD100k as the target domain for evaluation.
> Our method still outperforms other methods thanks to our more sufficient large domain style variance guaranteed by the feature noise term $x^2 \sigma_\alpha^2$.
> Besides, it is usually non-trivial to train a detection model on multiple datasets because of the inconsistent label space.
> Thus our method has the great advantage of without any assumption on the source domain.
>
> | C & F & K & S  | mAP50 |
> |---------------------|---------|
> | Baseline            | 30.1    |
> | Baseline + pAdaIN   | 38.6    |
> | Baseline + Mixstyle | 41.1    |
> | Baseline + DSU      | 45.3    |
> | Baseline + NP       | **47.2**    |
>
>
> **Q4. A few missing references.**
>
> **A4.** Thanks for your helpful suggestion. We cited these papers in our revised paper.
>
> ----
>
> **References**
>
> *Xingang Pan, Ping Luo, Jianping Shi, and Xiaoou Tang. Two at once: Enhancing learning and generalization capacities via ibn-net. In ECCV, 2018.*
>
> *Nam, Hyeonseob, and Hyo-Eun Kim. Batch-instance normalization for adaptively style-invariant neural networks. In NeurIPS, 2018.*

---

> ### Author Response · Authors · 2022-11-18
> **Author Responses to Reviewer kYbi**
>
> ----
>
> We would like to sincerely thank you for your efforts and valuable comments to improve our work!
>
> Below we address your concerns.
>
> ----
>
> **Q1. Discussion and comparison to IBN and BIN methods.**
>
> **A1.** Thanks for your valuable suggestion.
>
> **First, we have already discussed various normalization-based methods in the related work section.**
>
>
> **Second, per your valuable suggestion, we take a further discussion on IBN (Pan et al., 2018) and BIN (Hyeonseob and Kim, 2018) methods.**
> Both IBN and BIN strategically combine the IN and BN layers to improve the learning and generalization capacities, through the learnable gate and careful integration, respectively.
> They target at preserving image content information and reducing disturbing image style information, which is fundamentally different from our normalization perturbation manner.
> The table below shows our NP method substantially outperforms BIN and IBN methods, thanks to our effective style synthesis, which enables the model to perceive diverse potential domains during training.
>
> | Method           | **C** | **F** | **B** |
> |----------------|-------|-------|-------|
> | Baseline       | 58.0  | 22.0  | 21.8  |
> | Baseline + BIN | 58.5  | 28.4  | 25.3  |
> | Baseline + IBN | 58.3  | 31.2  | 27.1  |
> | Baseline + NP  | **58.7**  | **44.0**  | **30.1**  |
>
> ----
>
> **Q2. Discussion and comparison to image-level data augmentation.**
>
> **A2.** Thanks for your valuable suggestion.
>
> **First, we have already discussed the reason our feature-level perturbation is better than the image-level in Section 4.2 of the main paper.**
> Let us reiterate here for your reference:
> >Note that image-level domain synthesis methods may destroy the content structures of the original images in the image generation procedure. Besides, NP trains deep models with numerous content-style combinations in the high-dimensional feature space, which is much more efficient and effective than the image-level methods, whose styles are deterministic, limited and that their style augmentation is only performed on the low-dimensional image space.
>
> Besides, the input image can be regarded as a feature map with the RGB color channels. Thus the image-level data augmentation can be treated as a special case of feature-level augmentation, but is performed in the low-dimensional RGB color channels, which is therefore less effective than our feature-level perturbation in the high-dimensional feature channels.
> We also have the comparison to data augmentation in Table 9 of the appendix.
>
> **Second, to further consolidate our effectiveness, we compare various methods under the condition of using or not using data augmentation.**
> The tables below show the photometric data augmentation (i.e., Color Jittering, GrayScale, Gaussian Blur and Solarize) slightly impairs the in-domain detection performance, but significantly improves the generalization performance on different target domains.
> Our method consistently performs best in all settings.
>
> | Without Data Augmentation | **Baseline** | **Mixstyle** | **DSU** | **NP** | **NP+** |
> |------|--------------|--------------|---------|--------|---------|
> | Cityscapes    | 58.0         | 57.7         | 58.5    | 58.7   | **58.8**    |
> | Foggy Cityscapes    | 22.0         | 30.1         | 34.1    | **44.0**   | 43.2    |
> | BDD100k    | 21.8         | 26.5         | 27.2    | **30.1**   | 29.9    |
>
>
> | With Data Augmentation | **Baseline** | **Mixstyle** | **DSU** | **NP** | **NP+** |
> |------|--------------|--------------|---------|--------|---------|
> | Cityscapes    | 57.2         | 57.5         | 58.2    | 57.6   | **58.3**    |
> | Foggy Cityscapes    | 35.5         | 40.2         | 43.1    | 45.2   | **46.3**    |
> | BDD100k    | 30.5         | 30.8         | 31.4    | 32.6   | **32.8**    |

---

### Official Review · Reviewer_9C5D · 2022-10-24

**Confidence:** 4
**Clarity, Quality, Novelty And Reproducibility:** See Strength And Weaknesses
**Correctness:** 4
**Technical Novelty And Significance:** 3
**Empirical Novelty And Significance:** 3
**Recommendation:** 6

**Strength And Weaknesses:**

Strength
- The proposed Normalization Perturbation (NP) method could synthesize various new domain styles in an extremely simple way.
- The proposed methods cover various potential unseen domain styles in real-world environments, e.g., nighttime and foggy. Enables training models on cityscape to generalize well on Foggy Cityscapes and BDD 100K.

Weakness
- I tend to think that the way of increasing model robustness proposed in this paper is not designed for the object detection task. This is because the approach is a style transformation of image-level features, but ignores box-level features.
- The classwise mAP is not listed in the paper, so it is not clear whether NP could enhance detectors to be robust to all categories.
- I question the value of this style synthesize method NP for application in object detectors, whether it can be added to a model that already has strong augment and still have large gains, for example, whether it can be added to the current SOTA detectors with strong augment, like dino or Yolo v6.
- Whether this approach will interfere with other data enhancement methods designed for detection tasks, such as InstaBoost. What ratio of NP should be added, if feasible?



**Summary Of The Paper:**

The paper explored the style-sensitive channels for effective style synthesis and proposed Normalization Perturbation for robust object detection. Their method generalizes well under some OOD detection datasets, like Foggy Cityscapes.

**Summary Of The Review:**

See Strength And Weaknesses

---

> ### Author Response · Authors · 2022-11-18
> **Author Responses to Reviewer 9C5D**
>
> ----
>
> **References**
>
> *Hao-Shu Fang, Jianhua Sun, Runzhong Wang, Minghao Gou, Yong-Lu Li, and Cewu Lu. Instaboost: Boosting instance segmentation via probability map guided copypasting. In CVPR, 2019.*
>
> *Li, Chuyi, Lulu Li, Hongliang Jiang, Kaiheng Weng, Yifei Geng, Liang Li, Zaidan Ke et al. YOLOv6: a single-stage object detection framework for industrial applications. In arXiv preprint arXiv:2209.02976, 2022.*

---

> ### Author Response · Authors · 2022-11-18
> **Author Responses to Reviewer 9C5D**
>
> ----
>
> We would like to sincerely thank you for your efforts and valuable comments to improve our work!
>
> Below we address your concerns.
>
> ----
>
>
> **Q1. NP method is not designed for the object detection task.**
>
> **A1.** Thanks for your valuable question.
>
> **First, yes, our method is not restricted to robust object detection.
> Scalability is one of our advances.**
> Our method is general and can be applied to various tasks and datasets.
> Our extensive experimental results in the appendix shows that our method is also superior in the semantic segmentation and classification DG tasks and datasets.
>
> **Second, we identify that the ''Small Domain Style Variance'' and ''Large Context Diversity'' are the main problems limiting previous DG methods on robust object detection.**
> Thus, we mainly focus on solving such main problems for robust object detection by perturbing feature channel statistics.
> The improvement on box-level features does not coordinate with our main research goal and is out of our scope, but it is still a promising future research direction.
>
> ----
>
>
> **Q2. Classwise mAP.**
>
> **A2.** Thanks for your valuable suggestion. We present the class-wise mAP in the below tables for your reference. The model is trained on Cityscapes dataset and evaluated on Foggy Cityscaapes and BDD100k. Our method consistently improves the generalization performance on all classes.
>
>
> | Foggy Cityscapes     | **person** | **rider** | **car** | **truck** | **bus** | **train** | **motorcycle** | **bicycle** | **mAP** | **mAP50** |
> |-----|---|-----|----|----|-----|----|-----|-----|-------------|---------|
> |  Baseline | 16.9       | 18.2      | 22.6    | 7.8       | 15.5    | 3.9       | 7.8            | 15.9        | 13.6    | 22.0      |
> |         NP       | 26.5       | 31.8      | 40.6    | 17.4      | 33.6    | 12.4      | **15.7**           | 24.9        | 25.4    | 44.0      |
> |                    NP+      | **26.8**       | **33.6**      | **40.9**    | **18.4**      | **37.1**    | **14.8**      | 15.4           | **25.4**        | **26.6**    | **46.3**      |
>
> | BDD100k     | **person** | **rider** | **car** | **truck** | **bus** | **train** | **motorcycle** | **bicycle** | **mAP** | **mAP50** |
> |-----|---|-----|----|----|-----|----|-----|-----|-------------|---------|
> |  Baseline | 20.6       | 10.9      | 32.5    | 5.8       | 7.4     | 0.0       | 4.2            | 11.1        | 11.6    | 21.8      |
> |          NP       | 29.5       | 15.5      | 42.2    | 9.8       | 9.7     | 0.0       | 7.5            | 16.1        | 16.3    | 30.1      |
> |                    NP+      | **30.5**       | **16.4**      | **42.5**    | **10.6**      | **11.7**    | **0.1**       | **12.5**           | **17.0**        | **17.7**    | **32.8**      |
>
> ----
>
>
> **Q3. The NP method effectiveness when combined with InstaBoost or SOTA detectors.**
>
> **A3.** Thanks for your valuable suggestion.
>
> The below tables show the results of applying InstaBoost (Fang et al., 2019) on Faster R-CNN and applying our NP method on YOLOv6 (Chuyi et al., 2022).
>
> Instaboost can consistently improve the detection performance on all testing datasets thanks to its object-level augmentation, but still heavily suffers from the domain shifts problem.
> Instaboost introduces random jittering to objects, which is orthogonal to our method focusing on perturbing feature channel statistics.
> **Thus our NP method can be fittingly combined with Instaboost to further boost the performance on all testing domains.**
> Moreover, our method is also orthogonal to other popular data augmentation methods, \ie, Mosaic and Mixup.
>
> | Method                           | **C** | **F** | **B** |
> |--------------------------------|-------|-------|-------|
> | Faster R-CNN                   | 58.0  | 22.0  | 21.8  |
> | Faster R-CNN + NP              | 58.7  | 44.0  | 30.1  |
> | Faster R-CNN + InstaBoost      | 60.5  | 26.1  | 24.2  |
> | Faster R-CNN + NP + InstaBoost | **60.7**  | **46.8**  | **33.0**  |
>
> We further conduct experiments on YOLOv6-M.
> The powerful YOLOv6 model significantly improves the detection performance on all datasets.
> But the performance improvement is mainly derived from the stronger backbone, better data augmentation (Mosaic and Mixup), and advanced detection techniques, which are all specifically designed for the in-domain object detection.
> YOLOv6 still heavily suffers from the real-world domain shifts problem.
> **Our NP method can be effectively combined with YOLOv6 to further boost the performance on all datasets.**
>
> **In summary, our NP method can be fittingly combined with most advanced detection techniques to further boost the detection performance, *e.g.*, popular data augmentation techniques and SOTA detectors.**
>
>
>
> | Method                           | **C** | **F** | **B** |
> |--------------------------------|-------|-------|-------|
> | YOLOv6-M                       | 67.3  | 32.6  | 27.3  |
> | YOLOv6-M + NP                  | **67.4**  | **56.2**  | **35.7**  |

---

### Official Review · Reviewer_ZMVE · 2022-10-26

**Confidence:** 4
**Clarity, Quality, Novelty And Reproducibility:** The paper is well written and increme…
**Correctness:** 3
**Technical Novelty And Significance:** 2
**Empirical Novelty And Significance:** 2
**Recommendation:** 6

**Strength And Weaknesses:**

Strength
1. The domain adaption for object detection is very important research topic in real-world applications.
2. The motivation of the proposed method is clearly introduced and explained.
3. The proposed method is simple yet effective, demonstrated by extensive experiments.

Weakness
1. The authors claim that “shallow CNN layers which preserve more style information”. Are there any evidences to support this statement ?
2. How to make sure the proposed normalization perturbation can preserve it would not change the image content?


**Summary Of The Paper:**

The paper proposes a simple yet effective normalization perturbation method to synthesize various latent styles during model training so that model can perceive diverse potential domains and generalizes well even without observations of target domain data. Extensive experiments are conducted to demonstrate the effectiveness of the proposed method for object detection in real-world domain shift.

**Summary Of The Review:**

The motivation of the paper is with insight, the proposed method is simple yet effective but lack novelty.

---

> ### Author Response · Authors · 2022-11-18
> **Author Responses to Reviewer ZMVE**
>
> ----
>
> We would like to sincerely thank you for your efforts and valuable comments to improve our work!
>
> Below we address your concerns.
>
> ----
>
> **Q1. Evidence for ''shallow CNN layers preserve more style information''.**
>
> **A1.** Thanks for your valuable question.
> It is well known that the CNN networks encode the hierarchical features. The shallow layers correspond to corners, edge, color, texture, \etal, which mainly contribute to the image style.
> The deep layers are more semantic-specific for objects, which mainly contribute to the image content.
> **Such observations have already been investigated through extensive visualization and analysis in various classic works** (Zeiler and Fergus, 2014; Huang \& Belongie, 2017; Dumoulin et al., 2017).
> Mixstyle (Zhou et al., 2020) and IBN (Pan et al., 2018) also find and conclude that ''style information is preserved at the bottom layers of the CNN through the instance-level feature statistics''
>
> ----
>
>
> **Q2. How to make sure the proposed normalization perturbation can preserve it would not change the image content?**
>
> **A2.** Thanks for your insightful question.
>
> **First, we have never claimed that ''our method would not change the image content''.
> Instead, our claim is that ''our method maintains high content fidelity to original images''.**
>
> **Second, we empirically show that our method can often preserve well the image and feature spatial structures, through our better quantitative results in Table 3(a) and the better qualitative results in Figure 4 of the main paper.**
>
> **Third, previous works (Huang \& Belongie, 2017; Pan et al., 2018; Hyeonseob and Kim, 2018; Zhou et al., 2020) have also made the same observation that channel-wise feature normalization operation preserves well the image content**, and thus propose various feature normalization techniques based on such observation. Recently, ArtFlow (An et al., 2021) theoretically proves that the adaptive instance normalization in AdaIN is an unbiased style transfer module using the Bilinear Model (Tenenbaum and Freeman, 2000) framework, *i.e.*, AdaIN separates deep features into the content information (normalized feature maps) and the style information (feature channel statistics). Our NP method is a variant of AdaIN and preserves well the image content information, by keeping the normalized feature maps unchanged.
>
> ----
>
>
> **Q3. Incremental method novelty.**
>
> **A3.** Our method is simple but effective, insightful and novel, **evidenced by the formulation comparisons in Table 1 of the main paper and our better performance in extensive robust object detection experiments.**
>
> Our novelty and contribution are two folds.
>
> We perform the first attempt on investigating the real-world domain shifts for robust object detection and highlight the shortcomings of the existing domain synthesis approach. We contribute deeper insights on adapting existing classification DG methods to effectively address the object detection DG problem.
>
> Second, we propose a simple and effective domain generalization method for robust object detection **without any assumption on the inputs**. Our method is orthogonal to other methods and thus can benefit other methods, e.g., data augmentation, Mixstyle and DSU as well.
>
> ----
>
>
> **References**
>
> *Xun Huang and Serge Belongie. Arbitrary style transfer in real-time with adaptive instance normalization. In ICCV, 2017.*
>
> *Vincent Dumoulin, Jonathon Shlens, and Manjunath Kudlur. A learned representation for artistic style. In ICLR, 2017.*
>
> *Matthew D Zeiler and Rob Fergus. Visualizing and understanding convolutional networks. In ECCV, 2014.*
>
> *Kaiyang Zhou, Yongxin Yang, Yu Qiao, and Tao Xiang. Domain generalization with mixstyle. In ICLR, 2020c.*
>
>
> *Xingang Pan, Ping Luo, Jianping Shi, and Xiaoou Tang. Two at once: Enhancing learning and generalization capacities via ibn-net. In ECCV, 2018.*
>
> *Jie An, Siyu Huang, Yibing Song, Dejing Dou, Wei Liu, and Jiebo Luo. Artflow: Unbiased image style transfer via reversible neural flows. In CVPR, 2021.*
>
> *Nam, Hyeonseob, and Hyo-Eun Kim. Batch-instance normalization for adaptively style-invariant neural networks. In NeurIPS, 2018.*
>
> *Joshua B Tenenbaum and William T Freeman. Separating
> style and content with bilinear models. In Neural Computation, 2000.*

---

### Official Review · Reviewer_QrJD · 2022-10-27

**Confidence:** 3
**Correctness:** 2
**Technical Novelty And Significance:** 2
**Empirical Novelty And Significance:** 2
**Recommendation:** 6

**Clarity, Quality, Novelty And Reproducibility:**

The clarity and quality is good. Some extra explanations would be appreciated, for example how are $\Sigma_{\mu}(x)$ and $\Sigma_{\sigma}(x)$ computed in Table 2 and how are "Jigsaw" implemented in Table 5.

The details are enough to reproduce the work.

**Strength And Weaknesses:**

### Strength

* The paper focuses on object detection (and segmentation as well) which is a more practical and more difficult vision task comparing with classification.
* The proposed method does not require extra diversity in the training set or a combination of multiple training sets.
* The performance is pretty good on multiple datasets, and under multiple settings.

### Weaknesses
* Authors claim that Mixstyle and DSU do not work well since the detection training dataset does not have enough variance by default. But it maybe worth trying to add image augmentation to increase the variance of the training data.
* The proposed method seems to be restricted to autonomous driving datasets. Although they may have different style (e.g. weather/time of day) across different domains, the variance between them are still small. It's not clear how the proposed method works on some more complicated variances like object rotation, indoor/outdoor context switch.

**Summary Of The Paper:**

This paper focuses on domain generalization for object detection, especially for the application of autonomous driving. Authors claim that features from images in different domains have different statistics, thus propose to perturb the feature statistics during the training. Experiments are conducted on several autonomous driving datasets with different weather, time of day and synthetic/real sources.

**Summary Of The Review:**

To summarize, the scope of the application is limited to a single setting (most evaluations are done in autonomous driving datasets), and it's not thoroughly investigated that if more straightforward data augmentation can achieve similar robustness.

----
Thanks for authors' response and extra experiments. I change my rating to weak accept as my concerns have been addressed.

---

> ### Author Response · Authors · 2022-11-18
> **Author Responses to Reviewer QrJD**
>
> ----
>
> We would like to sincerely thank you for your efforts and valuable comments to improve our work!
>
> Below we address your concerns.
>
> ----
>
> **Q1. Method comparisons with data augmentation.**
>
> **A1.** Thanks for your valuable suggestion.
>
> **First, We have discussed the effects of data augmentation in Table 9 (c) of the appendix.**
>
> **Second, following your suggestions, we conduct more experiments by applying the photometric data augmentation**, *i.e.*, Color Jittering, GrayScale, Gaussian Blur and Solarize, on the baseline, Mixstyle, DSU, our NP and NP+ methods. The below table shows that the data augmentation usually impairs the in-domain detection performance (*i.e.*, on C, Cityscapes,) because
> such augmentation introduces explicit domain style inconsistency to the training images. Inspiringly however, the data augmentation consistently improves the generalization performance on target domains (*i.e.*, on F, Foggy Cityscapes and B, BDD100k), benefiting from the resulting larger domain style variance. **But our NP and NP+ methods still outperform other methods thanks to our more sufficient large domain style variance guaranteed by the feature noise term $x^2 \sigma_\alpha^2$.**
>
> | Without Data Augmentation | **Baseline** | **Mixstyle** | **DSU** | **NP** | **NP+** |
> |------|--------------|--------------|---------|--------|---------|
> | Cityscapes    | 58.0         | 57.7         | 58.5    | 58.7   | **58.8**    |
> | Foggy Cityscapes    | 22.0         | 30.1         | 34.1    | **44.0**   | 43.2    |
> | BDD100k    | 21.8         | 26.5         | 27.2    | **30.1**   | 29.9    |
>
>
> | With Data Augmentation | **Baseline** | **Mixstyle** | **DSU** | **NP** | **NP+** |
> |------|--------------|--------------|---------|--------|---------|
> | Cityscapes    | 57.2         | 57.5         | 58.2    | 57.6   | **58.3**    |
> | Foggy Cityscapes    | 35.5         | 40.2         | 43.1    | 45.2   | **46.3**    |
> | BDD100k    | 30.5         | 30.8         | 31.4    | 32.6   | **32.8**    |
>
>
> ----
>
>
> **Q2. The proposed method seems to be restricted to autonomous driving datasets?**
>
> **A2.** Thanks for your constructive suggestion.
>
> **First, we have already applied our method on other datasets and tasks with reported results in Table 10 (for the classification DG at PACS), Table 11 and Table 12 (for the segmentation DG on various datasets) of the appendix.** Although not specifically designed for classification and segmentation DG, our method still performs better or comparable to previous DG methods thanks to our diverse latent styles generated by the perturbation operation.
>
> **Second, we have already discussed our limitation in Section A.3 of the appendix, honestly admitting our limitation on handling the domain context discrepancy.** Such limitation is a common issue suffered by all the feature statistic perturbation based DG methods including ours. Your valuable suggestion inspires us to further explore the more complicated variances for robust object detection, *e.g.*, object rotation, indoor/outdoor context switch in our future work.
>
> **Third, our work mainly focuses on the autonomous driving scenario, which is of high academic significance, high practical relevance while challenging to computer vision researchers and industrial practitioners.** In the academic research community, a great number of large-scale datasets have been proposed specifically for autonomous driving, *e.g.*, KITTI, Cityscapes, BDD100k, nuScenes, Waymo Open Dataset, Mapillary and Apolloscape. The next CVPR 2023 even assigns a subject area specifically for autonomous driving. In the industrial community, a large number of autonomous
> driving Unicom companies have been established, *e.g.*, Waymo, Tesla, Momenta, Baidu’s Apollo and Pony.ai. Thus, rather than saying that our method is restricted to autonomous driving datasets, by contrast, our method demonstrates substantial potential in the ever-important area of autonomous driving, evidenced by our highly competitive performance on various autonomous driving tasks and datasets.
>
> ----
>
>
> **Q3. More experiment details.**
>
> **A3.** Thanks for your valuable suggestion.
> To compute $[\sum_\mu(x)]^2$ and $[\sum_\sigma(x)]^2$ for the dataset, we first shuffle the train set, and then sample a batch of $64$ images to extract their ResNet stage1 features, and finally we compute the variance of their feature channel statistics, *i.e.*, mean and standard deviation, respectively.
> We traverse all image batches of the train set and average all the computed variance values as the output.
> The Jiasaw setting is the same as in Cycle Confusion, *i.e.*, a 2x2 grid is sampled from each image and shuffled, and the detector has to predict the permutation of the tiles.

---

### Public Comment · ~Yuxi_Li2 · 2022-11-09
**Some questions**

Hi, I just came across this submission and admire the insightful motivation and analysis on the relationship between channel-wise statistics and generality of object detections. While I have some question about the technique in this paper, I appreciate if authors can provide more detailed discussion:

- The idea of perturbation on feature in terms of its global/local statistic is very similar to some works already designed for general data augmentation or DG in classification as below[1,2], I think it will be better to further discuss the advantage of NP over these methods

[1] Li, Pan, et al. "A simple feature augmentation for domain generalization." Proceedings of the IEEE/CVF International Conference on Computer Vision. 2021.

[2] Wang, Yulin, et al. "Implicit semantic data augmentation for deep networks." Advances in Neural Information Processing Systems 32 (2019).

- In the society of Domain Generalization and its application, it seems to be a consensus that cross-domain variance is due to the style, which is highly related to global statistic of image features, but adjustment in global statistics can not affect the local style, texture of geometry layout of a specific object. However, in some cases, the cross-domain difference can also lie in local and conceptual content, e.g. the concept "people" in a real photo and in a clipart manifest totally different texture or local layout, there are also some datasets of object detection considering this issue[3], I think it will be better to take this local variance into account for more generalizable object detection systems.

[3] Inoue et al. "Cross-Domain Weakly-Supervised Object Detection Through Progressive Domain Adaptation." Proceedings of the IEEE/CVF Conference on Computer Vision and Pattern Recognition. 2018.

---

> ### Author Response · Authors · 2022-11-18
> **Author Responses to Yuxi**
>
> ----
>
> Dear Yuxi,
>
> Thanks for your helpful comments to improve our work!
>
>
> **Q1. Discussion with SFA and ISDA.**
>
> **A1.** Thanks for your valuable suggestion.
> Actually, we have already discussed SFA (Li et al., 2021) in Section 2 of the main paper and presented more comparison details in Section A.2 and Table 8 of the appendix.
> ISDA (Wang et al., 2019) augments the training data semantically on top of deep models, which is fundamentally different from our normalization perturbation method applied in shallow CNN layers for efficient domain style synthesis.
>
> ----
>
> **Q2. Local variance for robust object detection.**
>
> **A2.** Thanks for your insightful suggestion.
>
> First, we totally agree that the consideration of image local content and style is useful for robust object detection.
> However, we identify that the ''Small Domain Style Variance'' and ''Large Context Diversity'' are the main problems limiting previous DG methods on robust object detection. Thus, we mainly focus on solving such main problems for robust object detection by perturbing feature channel statistics. The improvement on box-level features does not coordinate with our main research goal and is out of our scope, but it is still a promising future research direction.
>
> Second, our method is not restricted to robust object detection. Scalability is one of our advances.
> Our method is general and can be applied to various tasks and datasets.
> Our extensive experimental results in the appendix show that our method is also competitive in semantic segmentation and classification DG tasks.
>
> **References**
>
> *Pan Li, Da Li, Wei Li, Shaogang Gong, Yanwei Fu, and Timothy M Hospedales. A simple feature
> augmentation for domain generalization. In ICCV, 2021.*
>
> *Yulin Wang, Xuran Pan, Shiji Song, Hong Zhang, Gao Huang, and Cheng Wu. Implicit semantic
> data augmentation for deep networks. NeurIPS, 2019.*

---

### Author Response · Authors · 2022-11-18
**Author General Response**

----

**We would like to sincerely thank all reviewers for their efforts and valuable comments to improve our work! Reviewers appreciate our simple and
effective method, extensive experiments, and good writing.**

**We have uploaded a new version of our paper, revised based on reviewers’ valuable and helpful
comments. We highlight the revised parts in red color for better reference.**


----

---

> ### Comment · Reviewer_QrJD · 2022-11-27
> **Thanks for authors' response and extra experiments.**
>
> Thanks for authors' response and extra experiments. I change my rating to weak accept as my concerns have been addressed.

---

> > ### Author Response · Authors · 2022-11-27
> > **Very happy that we addressed your concerns!**
> >
> > Thanks for your positive feedback!
> >
> > We are delighted that we addressed your concerns.
> >
> > We also sincerely appreciate your efforts to improve our work.
> >
> > Your valuable and insightful comments inspire us to further explore practical solutions for challenging real-world applications.

---

### Decision · Program_Chairs · 2023-01-20

**Decision:**

Accept: poster

**Justification For Why Not Higher Score:**

Novelty seems a bit low.

**Justification For Why Not Lower Score:**

Simple but effective solution -- valuable to community.

**Metareview: Summary, Strengths And Weaknesses:**

Paper Summary:
Authors present a method of injecting random noise into feature channel statistics to improve domain generalization of object detectors. It does not require any new or diverse training data, is simply to implement, and yields significantly improved results over baselines.


Review Summary:

Pros:

- Important research topic (ZMVE)
- Clear motivation (ZMVE)
- Well written (kYbi)
- Focused on object detection which is more practical and difficult than classification (QrJD)
- Does not require extra diverse data (QrJD)
- Good performance in variety of settings (QrJD, ZMVE,9C5D,kYbi)
- Simple method (9C5D, kYbi)

Cons:
- Authors claim prior approaches do not work well. Perhaps authors can add data augmention (QrJD) -- authors have complied.
- What about datasets other than driving? (QrJD)
- Any evidence to support statement that style is in early CNN layers (ZMVE) -- authors have cited prior works
- How to ensure image content is not changed (ZMVE) -- authors have clarified statements. Image content may change, but the belief is not by much. Experimental evidence supports.
- Box level features ignored, shouldn’t be restricted to object detection only (9C5D) -- authors agree method generalizes to other settings and included experiments in appendix.
- Classwise mAP missing (9C5D) -- authors added
- Some missing references (kYbi) -- authors added
- Image perturbation / augmentation not compared (kYbi) -- authors added experiments to address.

AC Recommendation: Accept. Reviewers all lean to accept.

**Note From Pc:**

if the above contains the word "oral" or "spotlight" please see: "oral" presentation means -> notable-top-5% and "spotlight" means -> notable-top-25%. As stated in our emails, we are disassociating presentation type from AC recommendations